# GSK3α: An Important Paralog in Neurodegenerative Disorders and Cancer

**DOI:** 10.3390/biom10121683

**Published:** 2020-12-16

**Authors:** Octavio Silva-García, Ricarda Cortés-Vieyra, Francisco N. Mendoza-Ambrosio, Guillermo Ramírez-Galicia, Víctor M. Baizabal-Aguirre

**Affiliations:** 1Departamento de Química Teórica, Universidad del Papaloapan, Oaxaca 68301, Mexico; francisco.nc07@gmail.com (F.N.M.-A.); memorgal@gmail.com (G.R.-G.); 2Centro de Investigación Biomédica de Michoacán, Instituto Mexicano del Seguro Social, Michoacán 58000, Mexico; cortesvieyra@gmail.com; 3Centro Multidisciplinario de Estudios en Biotecnología, Facultad de Medicina Veterinaria y Zootecnia, Universidad Michoacana de San Nicolás de Hidalgo, Michoacán 58893, Mexico

**Keywords:** GSK3, GSK3 structure, GSK3α, GSK3β, cancer, Alzheimer’s disease

## Abstract

The biological activity of the enzyme glycogen synthase kinase-3 (GSK3) is fulfilled by two paralogs named GSK3α and GSK3β, which possess both redundancy and specific functions. The upregulated activity of these proteins is linked to the development of disorders such as neurodegenerative disorders (ND) and cancer. Although various chemical inhibitors of these enzymes restore the brain functions in models of ND such as Alzheimer’s disease (AD), and reduce the proliferation and survival of cancer cells, the particular contribution of each paralog to these effects remains unclear as these molecules downregulate the activity of both paralogs with a similar efficacy. Moreover, given that GSK3 paralogs phosphorylate more than 100 substrates, the simultaneous inhibition of both enzymes has detrimental effects during long-term inhibition. Although the GSK3β kinase function has usually been taken as the global GSK3 activity, in the last few years, a growing interest in the study of GSK3α has emerged because several studies have recognized it as the main GSK3 paralog involved in a variety of diseases. This review summarizes the current biological evidence on the role of GSK3α in AD and various types of cancer. We also provide a discussion on some strategies that may lead to the design of the paralog-specific inhibition of GSK3α.

## 1. Introduction

Glycogen synthase kinase-3 (GSK3) is a family of Ser/Thr kinases that is evolutionarily conserved in all eukaryotes that have been analyzed to date. Its first member was identified in protein extracts from rabbit skeletal muscle as part of a metabolic pathway capable of repressing the activity of the enzyme glycogen synthase via phosphorylation, independent of the activities of cyclic-AMP-dependent protein kinase (GSK1) and phosphorylase kinase (GSK2) [1]. In addition to its metabolic roles, GSK3 can regulate cell division, differentiation, autophagy, immunity, and apoptosis. This versatility is due to the broad range of its substrates, which currently include more than 100 proteins with diverse functions, such as receptors, structural proteins, signaling molecules, and transcriptional factors, making GSK3 one of the busiest kinases in the cell [2,3]. An unconventional characteristic of this enzyme is its constitutive kinase activity. This activity comprises its general mechanism of action as a repressor of its substrates via phosphorylation in unstimulated cells, representing a “break” that prevents the activity of most of its substrates.

After the initial discovery of GSK3, a detailed analysis of its expression led to the identification of two independent proteins with identical kinase activity encoded in different genes, but with a highly conserved sequence identity. These two enzymes were related and defined as paralogs, named GSK3α (the first member) and GSK3β (the second member) [4]. Both enzymes share 98% of the amino acid identity within their kinase domain and 100% similarity, being able to phosphorylate the same substrates [3]. Although these GSK3 paralogs were initially confined to metabolic pathways, an interesting discovery showed GSK3α to be the same protein previously known as the activating factor A (FA) of the (Mg-ATP-)dependent protein phosphatase and also related to the phosphorylation of neurofilaments of neuronal cells. Therefore, GSK3 family individuals were classified as multi-substrate kinases [5,6].

Since then, and according to bioinformatics prediction, the GSK3 paralogs have been demonstrated to have more than 500 substrates, making them the most active kinases in mammals [2]. These enzymes are involved in essentially every major process in the cell, including pathways such as apoptosis, insulin, phosphoinositide 3-kinase (PI3K), Wnt/β-catenin, Hedgehog, and Notch. The GSK3 proteins were previously thought to be equivalent because they show redundant functions as negative regulators of the same pathways when the other one is deleted [7,8]. In addition, although they are highly conserved among vertebrates, such as fish, reptiles, and mammals, an unexpected discovery has shown that in birds, the gene *GSK3A*, which encodes GSK3α, is absent. Moreover, in distant organisms such as the fruit fly, GSK3β but not GSK3α from a mammalian source can rescue embryos deficient in shaggy kinase (GSK3 homolog) [9].

One of the most important pieces of evidence showing differences between the GSK3 paralogs comes from the groundbreaking study by Hoeflich et al. [10]. This study revealed that GSK3α in KO mice was viable, while the deletion of GSK3β resulted in a lethal phenotype during mid-embryogenesis by promoting liver degeneration due to increased apoptosis of hepatocytes. This effect was related to the exposure of the gestating mice to pathogens, which in turn resulted in the release of tumor necrosis factor alfa (TNFα) and promoted hepatotoxicity. Interestingly, in a pathogen-free environment or during treatment of the gestating mice with anti-TNFα antibodies, the GSK3β KO mice embryos survived; however, they still died in the terminal stages of development or at birth. In these cases, death was due to cardioblast hyperproliferation inside the ventricular cavity [11,12]. This evidence showed that the GSK3 paralogs are not entirely equivalent and highlighted that GSK3β compensates for the functions of GSK3α; it was this evidence that established it as the dominant paralog [10].

More recently, it was found that GSK3β, but not GSK3α, regulates the innate immunity-associated expression of cytokines in macrophage-like Raw264.7 cells after Toll-like receptor (TLR) stimulation. This pro- and anti-inflammatory regulatory mechanism was associated with the activity of the nuclear factor kappa light chain enhancer of activated B cells (NF-κB), cyclic AMP response element binding (CREB), and the co-activator CREB binding protein (CBP) [13]. These and other observations have motivated the scientific community to focus on the GSK3β paralog and to assume that the functions of GSK3α are redundant or less representative; accordingly, they are often not even evaluated to discard any participation in analyzed cellular functions. We think that this vision underestimates the roles that GSK3α could be playing in the regulation of cellular processes and diseases, as recent evidence suggests specific functions for each paralog. Based on this idea, the following sections present evidence that supports the importance of GSK3α in neurodegenerative disorders (ND) and brain cancer, pancreatic ductal adenocarcinoma, prostate cancer, acute myeloid leukemia, multiple myeloma, and lung cancer. Moreover, we include recent experimental evidence that reveals how the dysfunction of GSK3α is related to pathological conditions in ND and cancer. We also address the current understanding of the structural differences between GSK3α and GSK3β that may allow the discovery of selective small molecule inhibitors for each paralog. Investigating the selective inhibition of GSK3α and GSK3β will be critical for elucidating the role that each paralog plays in ND and cancer.

## 2. The Relevance of GSK3α

In humans, GSK3α is the most abundant paralog expressed in almost every tissue according to datasets from the Human Protein Atlas [14]. Notably, its deletion in mice results in insulin sensitivity in some strains, accompanied by neurological alterations, while in testes, where the ratio of GSK3α/GSK3β is particularly high, it results in infertility. Interestingly, the active/inactive ratio of GSK3 paralogs has been proposed to be better evidence of the activity state than the total level [15], and remarkably, the evaluation of this relationship in several tissues suggests that GSK3α could be the most active paralog and the most sensitive to being inhibited by stimulation [16,17,18].

The analysis of GSK3β in the kidney or liver in conditional knockout (cKO) mice has shown that GSK3α can fully compensate for GSK3β loss to maintain the tissue function in postnatal individuals [19,20], suggesting that the lethality induced by the global deletion of GSK3β is related to the misregulation of cardioblast proliferation and differentiation during critical steps of embryo development [1,2]. In contrast to their wild type(WT) littermates, GSK3α KO mice develop normally up to 8 weeks of age, but they have a shorter life span, which is related to cardiac hypertrophy. Hence, in cardiac tissue from postnatal mice, GSK3β controls the proliferation and differentiation, while GSK3α regulates apoptosis and tissue remodeling in response to injury [21]. This has led to the conclusion that the coordinated regulation of both GSK3 paralogs is critical for the optimal function of the cardiac tissue [22,23]. In this context, the absence of GSK3α from cardiac tissue results in defective autophagy, leading to an impaired clearance of cellular debris, muscle contractile dysfunctions, and striking sarcopenia related to the premature death of GSK3α KO mice. This effect is associated with hyperactivation of the mammalian target of rapamycin complex 1 (mTORC1), which affects other tissues such as the testis, in which GSK3α, but not GSK3β, regulates germ cell spermatogenesis and the motility of spermatozoids [24]. The loss of the GSK3α/mTORC1 axis promotes exacerbated spermatogonial differentiation, leading to the depletion of germ cells [25]. Similar results have been reported in podocytes, in which GSK3α KO promotes cell death and impairs autophagy, which results in a detrimental glomerular function; in this case, however, no link to mTORC1 has been reported [26]. These effects could involve the differential regulation of gene expression because, while it is well-established that GSK3β and GSK3α have preference over some transcriptional factors, it has been reported that GSK3α is the main isoform that represses the activity of CREB, NF-κB, and mothers of decantaplegic homolog protein 3 and 4 (SMAD3/4) in neuronal cell cultures [27]. Its specific inhibition differentially regulates more than 200 genes in embryonic stem cells (ESC) [28] and acute myeloid leukemia (AML) cells [29]. 

As these transcriptional factors regulate the expression of multiple genes, such as cytokines, the differential regulation of gene expression represents a mechanism by which GSK3 regulates inflammation; however, the role of each paralog is not well-understood. Notably, although GSK3α activity is not related to the regulation of cytokine expression in macrophages upon the stimulation of TLR signaling [13], its deletion on myeloid cells from a high-fat diet mouse model promotes the polarization of macrophages to an M2 phenotype [19]. This leads to a reduction of the pro-inflammatory cytokine expression, which contributes to a reduction in the formation and progression of atherosclerotic lesions in comparison to WT or GSK3β myeloid-KO littermates [19]. In a different scenario, a recent study determined that GSK3α activity is negatively related to IL-10 and IL-12 expression, but positively related to the expression of IL-8 after stimulation with peptidoglycan (PGN) from *Staphylococcus aureus* or infection with *Staphylococcus aureus* of endothelial cells in a TLR2-dependent mechanism [17]. Other authors have found that GSK3α is the principal negative regulator of TNFα expression in neutrophils stimulated with lipopolysaccharide (LPS) [30]. This experimental evidence indicates the existence of specific immune responses regulated at the level of GSK3 paralogs that depend on the cell type and stimulus. In support of this theory, the enzyme IkappaB kinase ε (IKKε) promotes Akt activity by the specific downregulation of GSK3α, but not GSK3β, to induce Th17 cell maintenance and proliferation in response to IL-1 [31]. In summary, GSK3α regulates specific cellular components that are important in cellular mechanisms that have not yet been fully characterized.

## 3. General Aspects of GSK3α in Neurodegenerative Disorders and Cancer

In the context of neurodegenerative disorders such as Alzheimer’s disease (AD), the upregulated activity of GSK3α has been found to be involved in the proteolysis processing of the Aβ peptide precursor in its toxic forms of Aβ40 and Aβ42 [32,33]. This paralog has been identified as the preferential tau kinase, which promotes phosphorylation at the disease-related residues S396, S235, and T231 in various models of AD [6,33,34,35,36]. GSK3α upregulation is equally involved in cancer development and maintenance, where its expression and activity correlate positively with stage development and a poor prognosis in hepatocellular carcinoma, with up to 13-fold overactivity compared to normal cells [37]. In striking contrast, GSK3β expression in cancer cells is reduced in comparison to normal or pericancerous tissues, which correlates to clinical-pathological characteristics and a poor prognosis [38]. Similar results have been obtained in human cervical carcinoma [39], tumor cells from the thyroid gland [40], prostate carcinoma cells [41], and AML cells [29,42]. This relevant role of GSK3α inhibition as a target against cancer has been strengthened by other strategies. For example, from the screening of a library composed of 3517 chemical compounds, the inhibition of GSK3 was identified as the main candidate for inducing the differentiation and apoptosis of AML cells. Interestingly, by testing a library of shRNA constructed against 1000 human proteins, it was possible to identify the inhibition of GSK3α over GSK3β as the principal cause for the effects observed on AML cells [43].

The preferential activity of GSK3α over GSK3β in cancer cells has also been reported in pancreatic ductal adenocarcinoma (PDA), in which GSK3α, but not GSK3β, promotes activation of the canonical and non-canonical NF-κB pathways to promote cell viability and apoptosis resistance [44,45]. In p53-null colon cancer cells, GSK3α inhibition is able to re-establish apoptosis in chemotherapy-resistant cells after treatment [46]. It is clear that GSK3α regulates specific cellular functions and has emerged as an attractive target for developing pharmacologic therapies against AD, cancer, and other neurodegenerative disorders. There is an inverse relationship between cancer and AD, in which the activity state of the Wnt signaling pathway is an important factor [47,48]. Therefore, GSK3α would be a relevant target to evaluate given its involvement in both diseases and as a regulator of the Wnt pathway. However, there are also some issues that need to be overcome. 

## 4. GSK3α in Alzheimer’s Disease and Neurodegenerative Disorders

Alzheimer’s disease is a major cause of neurodegenerative dementia. It is characterized by the constant presence of two hallmark physiopathological injuries in the brain tissue: Senile plaques in the extracellular space formed by the polymerization of the Aβ peptides of 40 and 42 amino acids, and neurofibrillary tangles in the intracellular compartment of neuronal cells, comprising hyper-phosphorylated forms of tau protein. These lesions impair the proper neural function and ultimately lead to cell death [49]. Aβ peptides were shown to activate GSK3β, also known as tau protein kinase I (TPKI), in rat hippocampal neurons. Antisense oligonucleotides against TPKI prevented neuronal death [50] and LiCl, which is a GSK3 inhibitor, blocked Aβ peptide synthesis, which highlights the relationship between GSK3 activity and the Aβ peptide and vice versa. Although a role for GSK3β was identified, the role of GSK3α was largely overlooked. Interestingly, LiCl has also been reported to increase, rather than decrease, Aβ peptide synthesis [51]. This correlates with reports that indicate an increase of Aβ peptide synthesis by GSK3β inhibition [32]. Because other GSK3 inhibitors have been shown to reduce Aβ peptide synthesis, and GSK3α-specific inhibition resembles this effect [32], this paralog is considered the most attractive target for the development of pharmacological therapy. Immunohistochemistry targeting of GSK3α/β in brain tissue from AD patients shows a co-localization signal with hyper-phosphorylated tau. Nevertheless, the inactive form phospho-Ser9-GSK3β was found to be co-localized in up to 80% of cells with hyper-phosphorylated tau, indicating that this paralog is not the main one responsible for tau phosphorylation [52]. 

In AD, the Aβ-peptides that compose the senile plaques are derived from proteolysis of the amyloid peptide precursor protein (APP) in a coordinated sequence catalyzed by aspartyl protease beta-secretase 1 and 2 (BACE1/2) to generate the C-99 precursor that is processed by the presenilin-dependent γ-secretase [53]. In CHO cells stably expressing recombinant APP, the attenuation of GSK3α expression leads to a reduction in Aβ40/42 peptide synthesis. Surprisingly, treatment with siRNA against GSK3β promotes synthesis of the Aβ40/42 peptide. This effect was also reproduced in naive mouse neuronal cells and cells stably expressing the Swedish APP mutant [32]. In this study, the pretreatment of cells with LiCl did not affect the relative expression of the APP, which indicates that the interference of Aβ synthesis occurs at the posttranscriptional step during APP processing. Moreover, in SH-SY5Y human neuroblastoma cells, transfection with siRNA against GSK3β, but not GSK3α, reduced the processing of APP to the intermediate C-99 fragment by reducing the BACE1 expression [54]. However, siRNA against GSK3α reduced Aβ40/42 peptide synthesis, indicating that the regulation of APP processing by each GSK3 paralog can take place at different steps and suggesting the regulation of GSK3α over presenilin-dependent γ-secretase (Figure 1). To the best of our knowledge, this theory has not been tested. 

In a screening array applied to find proteins capable of interacting with Aβ42, GSK3α emerged as the top candidate among more than 9000 proteins [33]. This interaction was later validated by surface plasmon resonance, thermophoresis, pull-down assays, and confocal microscopy in primary cultures of neuronal cells from mice. Interestingly, in an in vitro kinase assay with purified GSK3α and tau, the addition of Aβ42 peptides stimulated GSK3α-dependent phosphorylation of tau at Ser396, indicating that GSK3α is a link between the two pathological hallmarks found in the tissue of AD patients [33]. In transgenic mice expressing mutant APP PDAPP+/− [55] or APP and TAU PS19+/− PDAPP+/− [56] that were transfected with adenovirus-associated GSK3α/β shRNA, only GSK3α silencing prevented Aβ peptide synthesis and senile plaque deposition in mice of 11 months of age and improved memory skills in mice of 17 months of age [36]. In contrast, the GSK3β silencing showed a tendency to increase Aβ peptide synthesis, in agreement with previous data from CHO cells [32]. Taken together, these results indicate that GSK3α-specific inhibition could be beneficial in repressing APP processing to produce the Aβ peptide and consequently, may be used as a treatment for AD. In a contrasting report, Jaworski et al. [57] found that neither GSK3α nor GSK3β regulate APP processing in vivo in KO or neural cKO mice. In this study, the inhibition of either GSK3 paralog was conducted in germline models; however, data on GSK3 regulation in APP processing were obtained in models where the inhibition of GSK3 was conducted in differentiated cell cultures or postnatal mice. This suggests that the developmental stage at which GSK3 is inhibited in biological models could be an important factor, as diverse mouse models of AD show different degrees of pathology. In this context, an increase in the phospho-inhibited GSK3α/β isoforms, coincident with a spontaneous accumulation of Aβ peptide in Sgo^+/−^ mouse models that was Wnt pathway activation-dependent, was observed during the late onset of AD [58]. This Aβ accumulation is in agreement with the findings reported by Phiel et al. and others [32,49] as GSK3β inhibition led to an increase in Aβ production. This fact highlights the need to prevent the simultaneous inactivation of GSK3α/β isoforms, in order to avoid the side effects observed when both are inhibited. These contrasting results related to GSK3α biological effects have also been reported in cardiomyocytes after myocardial infarction (MI), where KO germline models have demonstrated deleterious effects in cardiac functions [59], while cKO postnatal models have been reported to be protective [60].

The tau protein is part of the microtubule cytoskeleton in neuronal cell axons, forming a bond between tubulin heterodimers to stabilize the fibers [61]. The hyper-phosphorylation of tau leads to its disassembly from the microtubule fibers and the loss of axon integrity. The tau free monomeric subunits tend to self-aggregate and form the basic structure of the neurofibrillary tangles inside the neuronal cells that constitute a hallmark of all forms of tauopathies. The tauopathies are classified by the involvement of anatomic areas, cell types, isoforms, and the phosphorylation status of the protein present in pathological lesions in the brain. In this scenario, GSK3 is a relevant regulator of tau because it is the most active tau kinase, which is able to phosphorylate 26 of its 45 identified phosphorylatable residues [62]. In transgenic mice, the inhibition of both GSK3 paralogs can prevent tau misfolding and phosphorylation at Ser202, Thr205, Thr231, and Ser262 [36]. Nevertheless, the *GSK3A* gene in birds is absent and an analysis of the phosphorylation of tau in the brains of *House sparrow* and *Zebra finch* revealed that, in comparison to a mouse brain, the relative abundance of the disease-related phospho-tau-Ser396 is significantly reduced, while phosphorylation at Ser202 and Ser404 shows the same tendency, but without statistical significance [35]. Similar to the results in bird brains, GSK3α KO mice exhibited phospho-tau-Ser396 reduction compared with WT mice. Moreover, the reduction in phospho-tau-Ser396 was related to age because the analysis of an adult bird brain showed undetectable phosphorylation compared to an embryonic brain [35]. Similar results have been obtained in GSK3α KO mouse strains of the double-transgenic line GSK3αKOx/tau.P301L, showing that in GSK3α-deficient animals, the levels of phospho-tau-Ser396/404, Ser199, and Thr231 are diminished compared to WT or tau.P301L control mice [63]. One conclusion drawn from these data is that GSK3α and GSK3β can modulate tau phosphorylation in the same epitopes because GSK3α deletion leads to a reduction of phospho-tau species, but not to their elimination. Nevertheless, a comparison of the role of each isoform in tau phosphorylation in PS19+/−; PDAPP+/− mice brains revealed that the attenuation of GSK3α expression preferentially reduces phospho-Thr231 and phospho-Ser262 levels compared to GSK3β silencing. Although the efficiency of silencing of each isoform may be related to the preferential action of GSK3α on tau, additional evidence suggests that GSK3α could be mainly related to a pathological phospho-tau increase. For example, in a different approach using a cell-based alpha screen assay, the overexpression of 352 human kinases in the neuroblastoma cells SK-N-AS showed that GSK3α is the top kinase that preferentially promotes the phosphorylation of tau at Ser396/404, Thr231, Ser235, and Ser202 [34]. The same study reported that GSK3α, but not GSK3β, preferentially promotes the synthesis and interaction with the pathological structure of the Aβ peptide that in turn stimulates GSK3α phosphorylation of tau at Ser396. Collectively, these results highlight the function of GSK3α as an important link between the two hallmarks of AD, disclose its prominent role, and point out its specific inhibition, emphasizing the need for a therapy based on paralog selection.

In addition to its participation in tauopathies, GSK3α is important in the regulation of synaptic plasticity. Optimal brain functionality relies on the capacity of neurons to build physical connections that enable intercellular communication and the development of neuronal networks [64]. In this context, both GSK3 paralogs are needed for axon formation in neuronal cells, which is a critical step for establishing the synapsis [65]. To evaluate the synapsis, the long-term depression/long-term potentiation (LTD/LTP) assay is often used. Interestingly, by using chemical inhibition screening to identify kinases involved in the regulation of LTD, only GSK3 out of 58 Ser/Thr kinases tested was found to be involved and it was shown to be related to the N-methyl-D-aspartate (NMDA) receptor [66]. In this study, the incubation of brain slices from the hippocampus with three structure-unrelated GSK3 inhibitors blocked the induction of LTD. 

Since the chemical inhibition cannot distinguish between paralogs, and to rule out an off-target effect, the characterization of synapsis plasticity in knock-in mice expressing constitutive active mutants of GSK3 paralogs determined that GSK3α, but not GSK3β, preserved the activity and repressed the proper LTD/LTP balance in mice brain samples [67], suggesting that synapsis functionality requires dynamic changes to the activity state of GSK3α. In a different scenario, short-term GSK3 inhibition with BIO or CH98 in neuronal cell cultures prevented changes in LTD due to plasticity repression of dendritic spine remodeling. Interestingly, GSK3α, but not GSK3β, silencing resembled the BIO and CH98 effect on spine remodeling, identifying GSK3α as an essential regulator of the dendritic morphology and neuronal structure plasticity [68]. In another approach, the expression of the gain of function mutant GSK3αS21A, but not GSK3βS9A, repressed the consolidation of neuronal plasticity by preventing the LTP/LTD between the Schaffer collateral to CA1 synapses in the hippocampus of mice [67]. In this study, the effect of irresponsive synapses may be related to synaptic shrinkage, as it has been reported that GSK3α, but not GSK3β, affected synaptic plasticity by regulating actin polymerization after chemical-LTD (chLTD) in DIV18 hippocampal neurons [68]. Additional evidence supporting the dominant role of GSK3α in LTD was recently found by Draffin et al. [69], who revealed that GSK3α was required to induce LTD in hippocampal neuronal cells. Interestingly, during LTD, GSK3α was anchored to the dendritic spines via tau, which suggests that GSK3α/tau axis malfunction is a relevant step in tau-related pathologies.

## 5. GSK3 in Cancer

The activity of several pathways, such as Wnt/β-catenin and PI3K, is essential for different types of cancer growth and survival. As GSK3 activity is a negative regulator of these pathways, its function was thought to be a growth repressor. Nevertheless, many studies support the opposite notion, as it has been reported that GSK3 inhibition with chemicals exhibits anti-proliferative and pro-apoptotic activities in various types of cancer (reviewed by Maqbool et al. [70]). Since small molecule inhibitors cannot discriminate specific functions of GSK3 paralogs, it is not clear whether these enzymes can regulate the proliferation, survival, or apoptosis of cancer cells, in a cooperative or independent manner. For these reasons and because recent evidence suggests that the two GSK3 paralogs differentially regulate the survival of several types of cancer, such as lung cancer, AML, glioma, pancreatic ductal adenocarcinoma, and prostate cancer cells, an integral view of GSK3 paralog functions would help to develop specialized therapies. In this field, a role for GSK3β has been established and excellent reviews have described and summarized its functions [71]. The following sections emphasize the prominent role that GSK3α plays in different types of cancer (Table 1).

### 5.1. GSK3α in Brain Cancer

The two GSK3 paralogs are differentially expressed in the brain and have been reported to regulate redundant and specific functions in physiological and disease conditions. Glioblastoma is an aggressive and malignant tumor development of the central nervous system, and several studies have identified that small molecule-GSK3 inhibition promotes apoptosis and cell cycle arrest of glioblastoma cells [72]. A recent report indicates that transfection with shRNA targeting GSK3α in U87 and A172 glioma cells upregulates GSK3β activity and results in increased levels of c-myc, pERK1/2 (Thr202/Thr204), and cyclin D1 [73]. Interestingly, in GSK3α-deficient cells, β-catenin is not stabilized; however, its nuclear levels are enhanced, in agreement with reports that suggest a negative role for GSK3α as a nuclear β-catenin regulator in a mechanism independent of induced degradation [74].

An aberrant expression of heterogeneous nuclear ribonucleoprotein A1 (hnRNPA1) has been found in various types of cancer and is associated with splicing regulation to promote apoptosis resistance and survival [75]. The KD of GSK3α in U87 glioma cells promotes hnRNPA1 stability and mRNA synthesis to upregulate splice variant expression of the anti-apoptotic proteins BIN-1 and Mcl-1 with the parallel repression of RON, while in GSK3β inhibition, the opposite effect occurs. The KD of GSK3α also promotes the expression of survival genes such as BCL-XL, as well as survival and cell cycle regulators cyclin B1, D1, E1, and E2. These effects were related to the increase in the activity state of p70S6K, in keeping with reports that highlight a specific relation between GSK3α KD and the upregulation of p70S6K in cardiomyocytes [76] and TH17 cells [31]. These data indicate that, in malignant glioma cells, the specific inhibition of GSK3β and the maintenance of GSK3α activity would be beneficial in repressing cell proliferation, confirming that GSK3 paralogs antagonistically regulate glioma cell survival and proliferation.

In contrast to these findings, a report by Carter et al. [77] shows that the inhibition of GSK3 activity with AR-A014418 in the neuroblastoma cell lines NGP and SH-5Y-SY reduces proliferation and induces apoptosis. The inhibitor AR-A014418 primarily downregulates phospho-GSK3α-Tyr279. In vitro assays show that when the media containing the inhibitor is replaced with regular media, the cells recover the proliferation potential and the phospho-GSK3α-Tyr279 level is restored, which suggests that GSK3α inhibition, but not GSK3β inhibition, is involved in the AR-A014418 induction of apoptosis. These antagonistic effects of GSK3α inhibition on glioma or neuroblastoma cell viability indicate the need to understand the cellular background to identify the inhibition of GSK3α as a therapeutic candidate against cancer of the nervous system.

### 5.2. GSK3α in Pancreatic Ductal Adenocarcinoma

Pancreatic ductal adenocarcinoma (PDA) is defined as the most common malignant growth of the epithelial duct cells of the pancreas. It is a highly lethal and difficult disease to treat, in part because of its resistance to pro-apoptotic stimuli, including TNFα and the TNFα-related apoptosis-inducing ligand (TRAIL). In this context, GSK3β has been specifically identified as a negative regulator of TNFα-induced apoptosis in PDA, and GSK3 inhibitors have been reported to reduce PDA progression and apoptosis resistance; however, the contribution of GSK3α is not fully understood. By inducing the gene silencing of each GSK3 isoform, Zhang et al. [78] found that independent GSK3 silencing in Panc04.03 cells enhances the sensibility to apoptotic stimulation by TRAIL, with an apparent tendency to be more potent for GSK3α than for GSK3β. Significantly, the apoptosis resistance against TRAIL stimulation involves the activation of an anti-apoptotic response that is regulated in part by the preservation of NF-κB and by IKK activity promoted by GSK3α. Remarkably, GSK3α mainly regulates the NF-κB response in Kras mutations harboring Panc-1 and MiaPaCa-2 PDA cells, where the GSK3 inhibitor AR-A014418 decreases its proliferation. Furthermore, the KD of GSK3α with shRNA reduces the viability and colony size in these cells more preferentially than GSK3β silencing. Notably, the inhibition of GSK3 with AR-A014418 also correlated with a decreased phospho-activation of IKKαβ (Ser176/180) and NF-κB p65 (Ser536); such activity is critical to the promotion of cell proliferation and survival. In this context, IKKα/β is downstream-regulated by the activity of transforming growth factor beta-activated kinase 1 (TAK1) and the specific inhibition of GSK3α, but not GSK3β, reduces the TAK1 abundance. AR-A014418 represses TAK1 dimer formation with TAB1 (active form) in Panc-1 and MiaPaCa-2 cells, and this result suggests that the inhibitory effect on PDA cell proliferation induced by AR-A014418 is primarily due to the loss of GSK3α, rather than GSK3β, activity. Moreover, GSK3α inhibition results in a reduced activation of NF-κB via the increase of p65 phospho-Ser536 and apoptosis via Caspase-3 activation [44]. AR-A014418 represses neuroblastoma cell proliferation by similar GSK3α preferential inhibition related to increased apoptosis [77]. This could be a key observation because it is estimated that Kras mutations are present in approximately 30% of human cancers where GSK3 activity is a requirement for the survival and proliferation of Kras-dependent tumors [79], as GSK3α is the link between Kras promotion of NF-κB pathway activity and the promotion of cancer cell viability. Its inhibition could represent new avenues to develop effective therapies.

Other evidence supporting the importance of GSK3α on PDA, reported by Bang et al. [44], shows that GSK3α, but not GSK3β, regulates the non-canonical NF-κB activity by promoting the processing of p100 to p52 in the nucleus. However, in Panc04.03 cells that also harbor Kras mutations, the KD of GSK3α sensitizes the cells to apoptosis induced by TRAIL at a similar potency to that of the KD of GSK3β. This fails when the stimulus is TNFα, suggesting that a GSK3α/NF-κB pathway could be more important to promoting cell growth, while the GSK3β/NF-κB pathway mainly promotes apoptotic resistance induced by TNFα in Panc04.03 cells. This evidence is in line with the findings regarding other tissues, such as the embryonic liver, where the degeneration triggered by TNFα-induced apoptosis is a result of the deletion of GSK3β, but not GSK3α [11].

### 5.3. GSK3α in Prostate Cancer

Both GSK3 paralogs are upregulated in about 30% of prostate tumor cells [80]; however, GSK3α expression shows a stronger correlation in low Gleason score tumors and is mainly associated with the promotion of cell proliferation in response to androgen receptor (AR) transcriptional activity [80], while GSK3β is mainly expressed on high Gleason score tumors and promotes proliferation in AR-independent mechanisms [81]. Transfection with shRNA targeting GSK3α in PC3, DU145, and LNCaP prostate cancer cells represses proliferation, survival, and colony formation in cell cultures and the tumor expansion of xenografts in athymic nude mice. Interestingly, transfection with shRNA targeting GSK3β represses the expression of Snail and MMP-9 (markers of migration and invasion), whereas GSK3α-deficient cells show a lower expression of Ki67 and enhanced detection of cleaved caspase-3 and caspase-9, which correlates with more intense TUNEL assay staining [82].

According to cell motility and micro-invasion assays, the silencing of GSK3α rather than GSK3β mainly represses the motility of cells in cell cultures. This effect was reproduced in a lung colonization assay by monitoring the invasion potential of PC3 cells in nude mice, resulting in a significant reduction of the tumor volume and weight of GSK3α-deficient cells [82]. In 22Rv1 and LNCaP cells, transfection with shRNA targeting GSK3α represses the cell colony number and AR transcriptional activity, as in PC3 cells. Additionally, in this system, the acute inhibition of GSK3α leads to a repression of AR binding to DNA without affecting its expression, while GSK3β inhibition promotes NF-κB pathway activity via an increased expression of RelB to promote cell survival [83]. This effect was also observed in ESCs, in which the specific inhibition of GSK3α leads to the generation of cell lines that spontaneously become apoptotic [28] after several passages, indicating that the acute or long-term inhibition of GSK3α shows differential cell behavior in differentiated and embryonic cell cultures. The mechanisms responsible for these observations are currently unknown. 

### 5.4. GSK3α in Acute Myeloid Leukemia

AML is a hematopoietic malignancy defined by dysregulation of the myeloid progenitor’s proliferation and disrupted differentiation. The role of the Wnt/β-catenin pathway is associated with the positive regulation of its development, and several of its inhibitors have been found to repress it [84]. As GSK3 paralogs are negative regulators of the Wnt/β-catenin pathway, its inhibition promotes the onset of AML from normal hematopoietic stem cells [85]. Nevertheless, several reports indicate that in already malignant cells from AML, GSK3 inhibition promotes cell differentiation, reduces its proliferation, and restores normal functions [86], without affecting bone marrow cells [87]. Therefore, GSK3 inhibition emerges as an attractive therapeutic candidate in AML. Although GSK3α deletion is not enough to initiate pre-neoplastic properties of hematopoietic stem cells, it contributes to the development of the disease by promoting aberrant metabolic activity of myelodysplastic syndrome cells [87].

In an interesting approach to defining potential candidates for the treatment of AML, Banerji et al. [43] used the intersection of pharmacological and genetic screens to identify GSK3α as the top candidate in both strategies. From a panel of 3517 chemical compounds, several GSK3 inhibitors were identified as myeloid differentiation promoters in the HL-60 cell line. Furthermore, in a second approach comprising 5036 shRNA targeting 1000 genes, the shRNA’s targeting of GSK3α emerged as the top candidates’ inducer of AML differentiation in HL-60 and U937 cells. Notably, when GSK3α or GSK3β were independently silenced to evaluate their relative contributions to AML differentiation, it was confirmed that GSK3α silencing has a more significant effect on cell differentiation in the four AML cell lines MOLM-14, U937, HL-60, and THP-1 [43]. The GSK3 inhibitors BRD0705 in U937 cells [29] and GS87 in HL-60, U937, THP-1, and NB4 AML cells promote differentiation and repress the proliferation rate [88], while the inhibition of both paralogs results in deleterious effects (Figure 2). It is important to note that both molecules show an increased selectivity for GSK3α, rather than for GSK3β, without any increase in Wnt/β-catenin pathway activity. This indicates that the specific inhibition of GSK3α is a good therapeutic strategy for ameliorating AML symptoms. 

### 5.5. GSK3α in Multiple Myeloma

Multiple myeloma (MM) is the uncontrolled proliferation of plasma cells that are clonally expanded from bone marrow progenitors. Primary MM cell lines exhibit a relative increase in the expression of GSK3α with respect to GSK3β when compared to normal peripheral blood mononuclear cells; such a preferential GSK3α expression effect is more evident in the MM cell lines U-266, OPM-2, RPMI8226, and INA-6. This GSK3α expression phenotype correlates with an enhanced activity, as the relative abundance of phospho-Tyr279-GSK3α is higher than phospho-Ser21-GSK3α and phospho-Tyr216-GSK3β/phospho-Ser9-GSK3β, respectively. The treatment of RPMI8226 and U-266 cells with the GSK3 inhibitor SB216763 promotes intrinsic apoptosis pathway activation characterized by the expression of the pro-apoptotic proteins Smac/DIABLO and Poly (ADP ribose) polymerase (PARP) cleavage. Moreover, MM cells treated with low concentrations of SB216763 mainly promote GSK3α inactivation, as evidenced by a stronger reduction of phospho-Tyr279-GSK3α than phospho-Tyr216-GSK3β. The transfection of MM cells with shRNA targeting GSK3α, but not GSK3β, sensitizes these cells to the cytotoxic effects induced by the proteasome inhibitor bortezomib [92]. These data suggest that both GSK3 paralogs could be involved in MM cell proliferation; however, GSK3α seems to have a prominent role and therefore emerges as an attractive target for MM therapy.

### 5.6. GSK3α in Lung Cancer

A recent report by Sin-Aye et al. [93] found that in 1760 lung cancer patients, GSK3α overexpression was a marker of a poor prognosis. The increased expression pattern of *GSK3A* mRNA from these samples correlated with CREB1, indicating a possible relationship. The analysis of the H1993 and H1437 lung cancer cell lines demonstrated that the transfection with shRNA against GSK3α or CREB, reduced cell viability and increased apoptosis. This result was reproduced in tumor growth xenograft assays, in which GSK3α-deficient H1993 cells inoculated in nude mice showed significantly less proliferation compared with H1993-WT cells. In this study, a positive GSK3α-dependent expression mechanism of cyclins A2, B1, D1, and E2 was uncovered and found to be correlated with the reduced proliferation of lung cancer cells via GSK3α inhibition. 

GSK3α silencing in the KRAS-WT cells H1993 and H1437 resulted in less proliferation than in the cells of KRAS mutants H1734 and A549 [93]. This is an important observation because GSK3α seems to be a link between mutant KRAS and NF-kB pathway activation that promotes apoptosis resistance in PDA cells [44]. However, the KRAS/GSK3α/NF-κB axis has not been tested in a lung cancer environment. Furthermore, the promoter region of the *GSK3A* gene possesses a CREB binding site; hence, GSK3α is a target of CREB transcriptional activity [93]. As CREB is related to numerous cell regulation mechanisms, the role of the GSK3α/CREB axis in other systems is an interesting mechanism to explore. These results show that, in lung cancer, GSK3α inhibition is beneficial for repressing cell viability and enhancing apoptosis, highlighting the need for a selective and specific GSK3α-inhibition-based therapy with chemical compounds.

**Table 1 biomolecules-10-01683-t001:** Function of GSK3α in various types of cancer.

Cancer Type	GSK3α Biological Effect	References
Hepatocellular carcinoma	Overactive	[37]
Cervical carcinoma	Overactive	[39]
Thyroid tumor cells	Overactive	[40]
Prostate carcinoma	(1) Overactive (2) Associated with the androgen receptor transcriptional activity (3) KD represses proliferation and Ki67 expression	[41,82,83]
Acute myeloid leukemia	(1) Overactive (2) KD or chemical inhibition induces cell differentiation	[29,42]
Glioblastoma	Its activity represses cell proliferation	[73]
Neuroblastoma	Chemical inhibition with AR-A014418 downregulates p-Tyr-279 and induces apoptosis	[77]
Pancreatic ductal adenocarcinoma	Its inhibition represses NF-κB activity and induces apoptosis via Kras mutations	[78,79]
Multiple myeloma	(1) Levels of expression higher than GSK3β (2) KD sensitizes cells to cytotoxic effects triggered by Bortezomib	[92]
Lung cancer	(1) Its overexpression is a marker of a poor prognosis (2) Promotes the expression of cyclins A2, B1, D1, and E2	[93]

## 6. Targeting a Single GSK3 for Therapy

GSK3 inhibitors show efficient activity in cell-free assays, but they fail in animal models due to the lack of selectivity against other kinases, insolubility, or low cell permeation. This means that from more than 2000 GSK3 inhibitors reported so far, only a handful have moved to clinical trials. Lithium and valproic acid are the exception because they have been frequently used in clinical applications, despite the fact that they show a narrow therapeutic window and their use for long-term periods is well-documented to produce side effects such as nephrogenic diabetes insipidus and life-threating disorders, especially in the elderly; however, these secondary effects have not been directly associated with GSK3 inhibition [94,95].

Notably, all designed dual inhibitors against GSK3 paralogs in cell cultures promote the stabilization of β-catenin, which is the main cause of GSK3-based therapy failure. The main restriction in the development of a GSK3 paralog-specific inhibition therapy is the high level of similarity between the predicted structures of the enzymes, as they share 98% of the amino acid identity, with 100% similarity in the catalytic domain. Nevertheless, various reports have shown that the selective inhibition of GSK3 paralogs with small molecule inhibitors is possible. For example, Feng et al. [96] synthesized a series of organometallic ruthenium-based molecules from which the first GSK3α selective inhibitor Λ-OS1 was found to have a seven-fold higher selectivity when compared to the inhibitory effect on GSK3β. Following this discovery, molecules of a distinct chemical nature have been characterized as selective inhibitors and several of them like the EHT352, with a significant difference of 30-fold selectivity for GSK3α [97]. However, the discovery of GSK3α selective inhibitors has mainly occurred through the incidental characterization of molecules targeting other proteins, and more recently, through computational prediction modeling and a pharmacophore-ligand-based strategy given that the crystallized structure of the protein is unavailable. Analysis of the molecules that preferentially repress GSK3α has generated evidence of differences between the GSK3 paralogs that could be exploited for a specific therapy [29] (Table 2). The next sections summarize the most important structural differences observed between the paralogs that can be exploited to discover and design highly selective inhibitors. 

### 6.1. The Substrate Binding Domain

The GSK3 paralogs display an enhanced affinity for pre-phosphorylated substrates. However, this is not strictly needed, as the phosphorylation of non-primed substrates has been reported [98,99]. In GSK3 paralogs, the domain that recognizes the pre-phosphorylated substrate is formed by a triad of basic amino acids. Arg96 in GSK3β is crucial in this triad because cell expression of the mutant R96A results in a reduction of the kinase activity toward primed substrates, but not to unprimed ones [100]. Interestingly, the expression of the GSK3α equivalent mutant R159A leads to a repression of the kinase activity toward unprimed substrates and primed substrates, such as recombinant expressed tau peptide [100]. These results indicate that the integrity of the phosphate-binding pocket in the pre-phosphorylating recognition domain of GSK3α is significantly more important than that of GSK3β, keeping its kinase activity toward unprimed substrates. From a different perspective, the interaction between the phosphorylated Ser21 present in the N-terminal domain of GSK3α with the pre-phosphorylated substrate pocket and the extended Gly domain with the rest of the protein could represent an approach for the rational design of paralog selective inhibitors because this feature is absent in GSK3β and, as far as we know, this has not yet been addressed. As the N-terminal end of GSK3α/β represses the kinase activity when Ser21 or Ser9 is phosphorylated by competing with the substrate for binding to the substrate recognition domain, the generation of mutant GSK3α-R159A should no longer be affected by the self-inhibition mechanism. This is because R159 must interact with the phosphate at Ser21 to downregulate the kinase activity [99]. The fact that the R159A mutant significantly loses its kinase activity suggests that the substrate-binding domain should be analyzed with the aim of designing selective inhibitors. Some of the GSK3 inhibitors, including manazmine A, are substrate competitive inhibitors. The concentrations of these molecules required to repress GSK3α have not been published, although manzamine A is a selective inhibitor of GSK3β, with no inhibition of GSK3α [101], which indicates that this domain could be a good region of the enzyme for designing paralog-specific inhibitors. 

### 6.2. The Glycine-Rich Loop Domain

The glycine-rich multifunctional loop (P-loop) domain in protein kinases, also known as the phosphate binding loop, is highly conserved [102]. The analysis of all the available crystallized structures of GSK3β reveals that this is a dynamic domain against all the GSK3-ATP-competitive inhibitors. This domain is formed in GSK3α by the residues 126 to 133, which are identical to 63 to 70 in GSK3β. However, dynamic simulation analyses reveal that this structure adopts differential conformations between the paralogs. These are observable between the GSK3β-WT (PDB 5KPM) and the mutant D133E (PDB 5T31), which simulates a change in the hinge domain of both GSK3 paralogs. Different spots present in the crystallized structures of the GSK3 D133E mutant were identified by Wagner et al. [29] and their presence can explain the selectivity to some inhibitors reported by others [96,103,104]. The mutant GSK3β-D133E shows the presence of a hydrophobic pocket formed by the amino acids Ile62, Val70, Tyr71, Gln72, and Ala83, but not present in GSK3β-WT. This pocket allows larger molecules to interact with this structure and confer paralog selectivity, as was confirmed by the crystalized structures of the GSK3 D133E mutant bound to BRD-0705 (PDB 5T31) and Λ-OS1 (PDB 3PUP). Both selective inhibitors interact with the same hydrophobic pocket (Figure 3). The pocket is not present in any of the analyzed structures of the GSK3β-WT counterpart, which in turn shows the presence of a hydrophobic pocket formed by Leu132—the amino acid adjacent to Asp133. This amino acid—Asp133—is the only difference between the two GSK3 paralogs inside the catalytic domain. 

### 6.3. The Hinge Domain

The GSK3α homologous proteins are present in a broad range of organisms, such as mammals, reptiles, the primitive fish *Latimeria chalumnae*, fungi, and the evolutionary-distant organism *Toxocara canis* [105,106]. The amoeba *Dictyostelium discoideum* only expresses one GSK3-like protein, named GSKA [107]. Analysis of the full amino acid sequence of GSKA shows a 78% identity match to human GSK3β and 76% to GSK3α. This makes it difficult to identify whether GSKA matches a particular GSK3 paralog from mammals because the amino acid sequences at the N- and C-terminals of the paralogs are significantly different, even in the same species. The human GSK3 paralogs share 98% identity within the catalytic domain, with only one amino acid difference located in the hinge backend region (Asp133 in GSK3β and Glu196 in GSK3α) equivalent position. The GSKA enzyme has Glu in this position, which suggests a more conserved sequence for mammalian GSK3α in this domain. Interestingly, *Arabidopsis thaliana* SHAGGY-related protein kinases (AtsK11 and AtsK12) also contain Glu in this equivalent position [108], which is also present in the canonical sequence of the Shaggy kinases from *Drosophila* and in the homologous GSK3 from the fungus *Ustilago maydis.* These observations are important for the following reasons: (1) The evolutionary history of GSK3 paralogs in eukaryotes is unresolved; (2) the primary sequence of all GSK3β orthologs reported to date shows Asp in the consensus sequence VL**D**YV, whereas in most of the GSK3α sequences, it is VL**E**YV; (3) this amino acid change in the GSK3 paralogs in humans has been shown to reorganize the H-bond network of the hinge domain that forms the bridge of the N-lobe to the C-lobe, which are the two major structures of the GSK3 paralogs. The network promotes various preferred conformations; and (4) this change may explain the paralog-selective inhibition of BRD0705, which shares binding properties with the Λ-OS1 inhibitor. This is supported by molecular dynamics simulations, docking analyses, and experimental evidence (see [29] for detailed information). 

### 6.4. Non-Conventional Pockets as Targets for Inhibitors

Although the ATP-binding pocket is frequently the main target of the GSK3 inhibitors, a computational study by Palomo et al. [109] shows the presence of seven cavities in GSK3β with a high potential to interact with some of its inhibitors. These proposed cavities correspond to the binding domains for ATP, the substrate, and the Axin/Fratide peptides; one of the last four pockets shows a high potential interaction with the known GSK3 inhibitor VP0.7. Another identified pocket has been proposed as an allosteric site where the non-ATP/substrate competitive inhibitor palinurin can bind and is localized in the N-terminal lobe [110]. This evidence indicates that in addition to the well-known ATP-binding pocket and substrate-binding domain, unconventional domains can be explored for the design of paralog-selective inhibitors against the GSK3 enzymes. An example is the non-ATP competitive and selective GSK3β inhibitor manzamine A that was isolated from the marine sponges *Haliclona* and *Acanthostrongylophora* [101]. 

### 6.5. GSK3α Selective Inhibitors

The IC_50_ value of 603 molecules has been determined for the two GSK3 paralogs from more than 2000 compounds reported as GSK3 inhibitors in public databases. Of these, 61 (~10%) show preferential selective inhibition for GSK3α and 21 (~3%) show a ten-fold or more preference for this isoform, with the top molecule 15b reported by Lo-Monte et al. displaying more than 90-fold selectivity [104]. The existence of molecules with potential for selective inhibition against one or the other GSK3 paralog indicates the presence of structural properties responsible for these preferential inhibitions. Hence, a computational analysis of the chemical fingerprinting may serve as the basis of the discovery or development of paralog-specific inhibitors and their binding mode, which can ultimately lead to the design of more specific inhibitors. We believe that the Quantitative Structure Analysis Relationship (QSAR) strategy is one of the most valuable tools for finding these features. Computational predictions may also be enriched with docking analysis performed against the GSK3 paralogs to find which pocket could serve as a discriminatory site interaction for the inhibition of each protein. Another possibility is the suitable chemical structure that a predicted paralog-specific inhibitor must display. Reports by Lo-Monte et al. [104,111] and Neumann et al. [89] show the development of oxadiazole-based compounds with selective inhibition potential for GSK3α. However, these compounds have a low solubility in aqueous solution. Optimization of the oxadiazole structure led to the synthesis of a compound with a 3.3-fold selectivity for GSK3α and promoted the differentiation of the AML cell lines HL-60 and NB4. This compound exhibited a selective reduction of the phospho-GSK3α-Tyr279, but not phospho-GSK3β-Tyr216, as this is an autocatalytic modification most often used to distinguish the activity of each paralog in a cellular context.

**Table 2 biomolecules-10-01683-t002:** Biological effects of GSK3 and GSK3 inhibition by selected compounds.

Molecule	GSK3α Inhibition, IC50 (nM)	GSK3β Inhibition, IC50 (nM)	Reported Biological Effect	References
BRD0705	66	515	Induces cell differentiation of AML cell lines and impairs colony formation of AML patient cells	[29]
EHT5372	7.44	221	Reduces phosphorylation of tau and Aβ production on recombinant expressing HEK293 cells	[97]
AZD2858	0.9	5	Reduces tau hyperphosphorylation in rat brains, and induces differentiation of mesenchymal progenitors to osteoblasts	[112] [113]
EHT1610	9.11	143	Reduces phosphorylation of tau and Aβ production on recombinant expressing HEK293 cells	[97]
8g	35	966	Unknown	[111]
8a	4	90	Unknown	[111]
8b	9	225	Unknown	[111]
27	42	140	Induces differentiation and impairs colony formation of AML, HL-60, and NB4 cells	[89]
26d	2	17	Unknown	[89]
14a	9	176	Unknown	[104]
15b	2	185	Unknown	[104]
14d	6	316	Unknown	[104]
G28_14	33	218	Induces differentiation and impairs colony formation of AML, HL-60, and NB4 cells	[114]
Λ-OS1	0.9	6	Unknown	[96]
Tivantinib	659	1865	Apoptosis of AML cells	[91]

## 7. Concluding Remarks

GSK3α and GSK3β, as we discussed above, regulate the activity of many substrates. Therefore, it is easy to anticipate that the simultaneous inhibition of both paralogs would deregulate multiple cellular processes and cause deleterious effects in several tissue-specific processes. For example, when both GSK3 proteins are downregulated with chemical inhibitors or by genetic manipulation during the embryonic development of mice, the phenotype observed is the hyperactivation of the Wnt/β-catenin pathway. This phenotype has lethal consequences due to the abnormal cell proliferation of undifferentiated stem cells during the embryonic development of mice, fruit flies, zebrafish, and *Xenopus laevis*. Even in postnatal stages, the simultaneous deletion of both kinases results in a lethal phenotype in mice that is characterized by a massive aberrant proliferation of cardiomyoblasts [115,116], probably by the repression of GSK3β, as this is the observed phenotype of the germline GSK3β KO mice.

Notably, if the inhibition is produced by chemical molecules that cannot discriminate between the two paralogs, mice develop cardiac hypertrophy that leads to deleterious effects during long-term periods of inhibition [117,118]. These observations have increased the interest in GSK3 therapy. For example, two of the GSK3 inhibitors that have progressed to phase I clinical trials are AZD2858 and AZD1080. These molecules are brain-permeable and display promising effects by downregulating tau hyper-phosphorylation and repairing synaptic dysfunction. Nevertheless, these are not paralog-selective inhibitors and have induced genotoxicity, gall bladder hyperplasia, and chronic cholecystitis side effects during in vivo testing in dogs and rats, which has prevented their further development [90]. The hypothesis that a paralog-specific therapy would be beneficial and safe remains to be tested; however, the question as to which paralog should be inhibited needs to be defined in each case. In mammalian cells, the GSK3β paralog is the main negative regulator of β-catenin [28], suggesting that the specific inhibition of GSK3α could be the safest target. This is supported by the absence of GSK3α in birds. Moreover, the phenotype of its deletion in mice results in less aggressive effects than GSK3β absence, such as insulin sensitivity. Intriguingly, Guezguez et al. [85] reported that when GSK3β or both GSK3 paralogs are conditionally deleted in hematopoietic stem cells, the observed phenotypes are consistent with hematopoietic dysplastic syndromes and ultimately lead to AML development. If GSK3α is selectively inhibited, the hematopoietic process is normal, without activation of the Wnt/β-catenin pathway [29,85].

Concerning AD, the inhibition of GSK3α results in a reduced synthesis of amyloid-β peptide aggregates, whereas GSK3β inhibition promotes it. Recently, the genetic manipulation of mice and cell cultures from various species led to an advance in the analysis of GSK3 paralogs. In this regard, Phiel et al. [32] reported that GSK3α silencing by siRNA assays reduces Aβ peptide levels in differentiated cell cultures [32]; however, more recent findings by Jaworski et al. [57] found no modifications of Aβ peptide levels in mice brain extracts from germline-strains GSK3α-KO, neuron-specific GSK3α, or GSK3β-KO. These results suggest that discrepancies may be linked to the tested model and that germline KO models of *GSK3A/B* genes and postnatal deletions may lead to different phenotypes, as reported for other systems [118].

The phosphorylation of either GSK3 paralog causes protein kinase activity inhibition. This feature makes GSK3 an atypical enzyme because it is permanently activated under resting conditions. GSK3 is one of the main suppressors of Wnt, Hedgehog, and Notch signaling pathways that control cell proliferation and differentiation, as well as stem cell maintenance. The inhibition of cell proliferation is achieved by GSK3 phosphorylation and targeting to degradation of the pro-oncogenic molecules β-catenin, c-Myc, and c-Jun. This GSK3 physiological function indicates that the inhibition of this enzyme is not a good therapeutic strategy for avoiding tumor progression. However, recent evidence suggests that the selective inhibition of GSK3α and GSK3β activity may shed light on the role of each paralog and also open new avenues for developing specific therapeutic strategies for each type of cancer. 

## Figures and Tables

**Figure 1 biomolecules-10-01683-f001:**
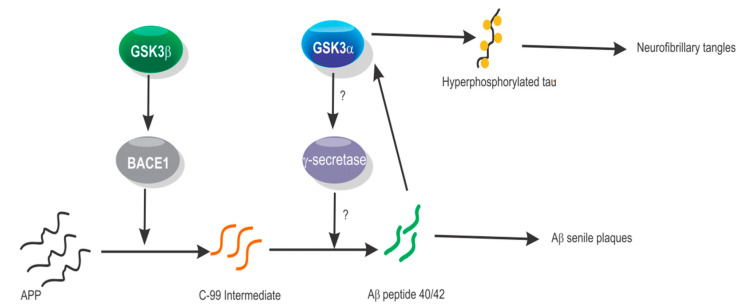
Proposed participation of each glycogen synthase kinase-3 (GSK3) paralog in the synthesis of the Aβ 40/42 peptide. The synthesis of the Aβ 40/42 peptide is the result of amyloid peptide precursor protein (APP) modification that leads to C-99 intermediate peptide formation. It seems that GSK3β regulates the activity of aspartyl protease beta-secretase (BACE1), while GSK3α modulates the activity of the presenilin-dependent γ-secretase (γ-secretase). The Aβ 40/42 peptide stimulates GSK3α to phosphorylate tau protein.

**Figure 2 biomolecules-10-01683-f002:**
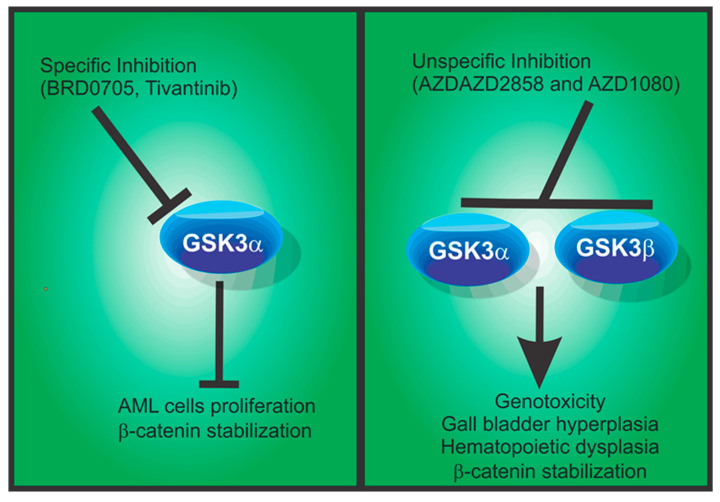
Specific inhibition of GSK3 may be beneficial in acute myeloid leukemia. Left panel: The selective inhibitors BRD-0705, Tivantinib, and others reported by Neuman et al. [89] promote the differentiation of acute myeloid leukemia (AML) cells without the stabilization of β-catenin and without deleterious effects [29,43,90,91]. Right panel: The inhibition of both GSK3 paralogs leads to deleterious effects and promotes β-catenin stabilization and cell proliferation [29,85].

**Figure 3 biomolecules-10-01683-f003:**
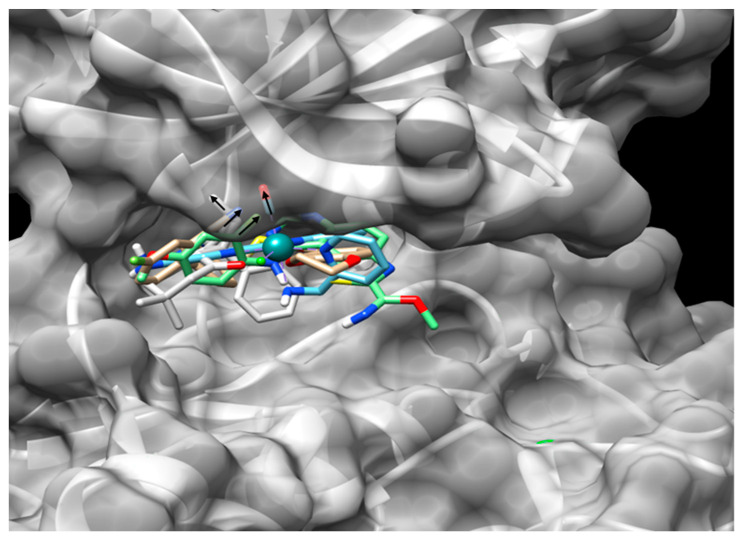
Molecular docking of GSK3α/β inhibitors. The ligands coupled to the GSK3α simulated protein GSK3β–D133E (white-gray surface) by Autodock Vina are color-coded as follows: 15b [104], golden; EHT1610 [97], green; BRD0705 [29], gray; and Λ-OS1 [96], blue. The arrows point to the region where the ligands interact with the cavity proposed by Wagner et al. [29]. Image generated with Autodock tools 1.5.6.

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
