# Peer review of "GSK3α: An Important Paralog in Neurodegenerative Disorders and Cancer"

_biomolecules, 2020, doi:10.3390/biom10121683_

Round 1

Reviewer 1 Report

The authors have written an interesting and timely review on GSK3α and its role in disease. On the whole the review has good coverage, is informative and should appeal to a good range of readers. The following are some suggestions for improvement.

1. The title and seems to reflect a specific focus on AD (a single neurodegenerative disease) and cancer (a range of malignancies). Given that GSK3α phosphorylates tau, its pathological role is really not limited to AD, but would include at least all forms of tauopathies. To balance the review a little, the authors may broaden their coverage to `neurodegenerative disorders’, and include discussions on some tau-related disorders.

2. An important aspect of neurological manifestations of diseases is impaired neurotransmission and excitation/inhibition imbalance. In this regard, the recent report by Draffin et al. in EMBO J. on GSK3α (but not GSK3β hippocampal NMDA receptor-dependent LTD could be included and discussed.

3. The intricacy of the functional interaction between GSK3α, tau and Aβ pathology in AD can be difficult to visualise. In this regard a simple diagram would aid the reader. Fig 1 is an attempt to do this but it is a little oversimplified in that tau is missing. Could this be improved?

4. The roles of the GSK isoforms in cancer is rather confusing and at times conflicting. A table summarizing the key findings in different cancers may help. Are there differential expression of the GSK isoforms (as well as amplifications, mutations etc.) in the different cancer tissues? This is an important point because selective targeting of GSK3α would not help if GSK3β is the pathological driver.

5. The title of Fig 3, ` Molecular docking of selective GSK3α inhibitors’ may be a little misleading as it is based on GSK3β-D133E. Are BRD-0705 (PDB 5T31) and Λ-OS1 (PDB 3PUP) really specific for GSK3α (ie. shown by biochemical assays)? A tabulated summary of all available GSK3α inhibitors along with their pharmacochemical properties would be desirable.

Reviewer 2 Report

Re: Octavio Silva-García et al., “GSK3a: An important paralog in Alzheimer disease and cancer” (review manuscript)

This manuscript intends to identify role differentiations in two isoforms of GSK3 (alpha and beta) with a focus on diseases, especially in Alzheimer’s disease and in cancers. The role differentiation is supported originally by KO mouse studies, then differential phenotypes in various KD/KO studies.

Overall, this manuscript is well written and informative, illustrating complex relationships between the two isoforms in different organs and disease backgrounds.

It may be helpful if authors can add or address following a few points.

(1) In another recent study, Rao et al. (Aging Cell, 2020, 19(10):e13221.) reported that in a genomic instability mouse model Sgo1 that accumulates cerebral amyloid-beta starting in middle age, both GSK3 alpha and beta were inhibited with inhibitory phosphorylation (S21, S9). The authors suggested that the GSK3 inhibition is a trigger for amyloid-beta accumulation in the middle age in the model that carries no mutation in APP or PSEN. Although their study did not provide evidence on differential regulations on GSK3 isoforms, a possible role of GSK3 in triggering amyloid-beta in middle age is worth mentioning or discussing.

(2) It may be helpful to remind readers of the involvement of age in AD development in AD mouse model studies. AD mouse models show different degrees of pathology development (e.g., https://www.alzforum.org/research-models/alzheimers-disease), but in many cases takes long time to develop AD pathology.

(3) From epidemiological studies, AD and cancer in human are suggested to have an inverse relationship (e.g., Papageorgakopoulos et al., Hell J Nucl Med. 2017 Sep-Dec;20 Suppl:45-57.). GSK3 isoform status analysis in this review also implicates non-uni-directional relationship of GSK3 isoforms and diseases. Can authors incorporate and discuss the AD/cancer relationship issue?
